# Inflammatory Biomarker Responses to Whole-Body Vibration in Subjects with Different Clinical Status: A Systematic Review

**DOI:** 10.3390/ijerph192214853

**Published:** 2022-11-11

**Authors:** Eloá Moreira-Marconi, Ygor Teixeira-Silva, Alexandre Gonçalves de Meirelles, Maria Eduarda de Souza Melo-Oliveira, Aline Cristina Gomes Santos, Aline Reis-Silva, Laisa Liane Paineiras-Domingos, Aderito Seixas, Carla da Fontoura Dionello, Danúbia da Cunha de Sá-Caputo, Mario Bernardo-Filho

**Affiliations:** 1Laboratório de Vibrações Mecânicas e Práticas Integrativas-LAVIMPI, Departamento de Biofísica e Biometria, Instituto de Biologia Roberto Alcântara Gomes and Policlínica Piquet Carneiro, Universidade do Estado do Rio de Janeiro, Rio de Janeiro 20950-003, Brazil; 2Programa de Pós-Graduação em Fisiopatologia Clínica e Experimental, Universidade do Estado do Rio de Janeiro, Rio de Janeiro 20551-030, Brazil; 3Departamento de Fisioterapia, Universidade Estacio de Sá, Rio de Janeiro 20771-900, Brazil; 4Programa de Pós-Graduação em Ciências Médicas, Universidade do Estado do Rio de Janeiro, Rio de Janeiro 20551-030, Brazil; 5Mestrado Profissional em Saúde, Medicina Laboratorial e Tecnologia Forense, Universidade do Estado do Rio de Janeiro, Rio de Janeiro 20950-003, Brazil; 6Departamento de Fisioterapia, Instituto Multidisciplinar de Reabilitação e Saúde, Universidade Federal da Bahia, Salvador 40110-060, Brazil; 7Escola Superior de Saúde Fernando Pessoa, 4249-004 Porto, Portugal; 8Departamento de Reumatologia, Universidade Federal do Rio de Janeiro, Rio de Janeiro 21941-590, Brazil

**Keywords:** inflammation, whole-body vibration exercise, physical and rehabilitation medicine

## Abstract

Background: Inflammation is considered to be a vital defense mechanism for health, acting as a protective response of the immune system through a satisfactory inflammatory biomarker response (IBR). IBR, as well as being beneficial to the organism, can be also responsible for a variety of chronic inflammatory diseases. Whole-body vibration (WBV) exercise is a type of physical exercise that can act on inflammation responses due its capacity for stimulating the sensory components that promote systemic responses. The objective of this study was to investigate the effects of WBV on IBR in different clinical status. Methods: This is a systematic review that includes randomized controlled trials (RCTs) on the effects of WBV exercise on IBR. The methodological quality, risk of bias, and level of evidence were rated. Results: Four RCTs met the selection criteria. The studies showed benefits associated with IBR (e.g., cytokines, adipokines, and C-reactive protein) in various clinical conditions, including healthy populations and some chronic diseases (such as obesity, aging disorders, and knee osteoarthritis), using several WBV protocols. Conclusions: WBV might be useful in the management of inflammatory conditions.

## 1. Introduction

Inflammation is considered to be a vital defense mechanism for health, acting as a protective response of the immune system [1,2]. It restores and defends physiological functions when homeostatic mechanisms are not sufficient, by replacing or suppressing incompatible homeostatic controls [1,2]. Thus, inflammatory mediators act by temporarily deactivating normal homeostasis, with a higher physiological priority placed on achieving a satisfactory inflammatory biomarker response (IBR) [1,2]. IBR (acute or chronic) is the protagonist in defending against life-threatening insults, infection, and injury [1,2,3], and can be triggered by several factors (endogenous or exogenous; infectious or non-infectious) [2,3]. IBR, as well as being beneficial to the organism, is also a potential mechanism for the onset of acute pathological states, which can become chronic when the acute form is not controlled, being responsible for a variety of chronic inflammatory diseases [1,2,3,4], such as osteoarthritis [5], obesity [6], type 2 diabetes [7], fibromyalgia [8], cardiovascular disease [9], and chronic obstructive pulmonary disease (COPD) [10]. IBR also can also be found in elderly populations, leading to inflammaging (i.e., chronic low-grade inflammation associated with aging) [11], as well in healthy individuals after physical exercise [12]. Therefore, it is important to identify physiological responses produced by inflammation in subjects with different clinical status, as the persistence or non-resolution of inflammation can cause changes in the defense mechanisms, leading to an excessive or abnormal IBR [4]. Thus, these mechanisms can be used for the development of anti-inflammatory therapies in chronic inflammatory diseases, including physical exercises.

Physical exercises have been suggested as a non-pharmacological intervention for the management of different clinical statuses. The biological responses to them depend on their type, duration, and intensity [13,14,15]. They can trigger important adaptations in the immune system [13,14,15]. Pereira et al. in 2012 [16], showed that a session of moderate-intensity exercise promoted an immediate positive IBR in individuals with heart failure. Deckx et al., in 2016 [17], found a reduction in the production of inflammatory mediators upon toll-like receptor stimulation and an increase in the immunoregulatory function of circulating plasmacytoid dendritic cells after a program of 12 weeks of combined endurance and resistance exercise. This is suggestive of a favorable impact of exercise on the underlying immune pathogenesis of individuals with multiple sclerosis. Whole-body vibration (WBV) exercise is a type of physical exercise that can act on inflammation responses [18,19] due to its capacity for stimulating the sensory components that promote systemic responses [20]. Another possible mechanism of action of WBV on IBR could be justified by the ability of physical exercise to regulate the immune system [21,22]. This could be related to its ability to reduce the number of senescent and exhausted T-cells and increase the proliferative capacity of T cells, decreasing the circulatory levels of inflammatory cytokines, improving the phagocytic activity of neutrophils, increasing the cytotoxic activity of natural killer cells, and increasing the length of leukocyte telomeres in aging humans [21,22]. Moreover, WBV is considered to be a physical exercise that promotes physiological alterations that are similar to those observed with traditional modalities of physical exercise, such as increased muscle performance [22,23].

WBV exercise is produced in individuals who are in contact with the base of a vibrating platform (VP) that transmits mechanical vibrations to them [24,25]. WBV exercise has been suggested as a non-pharmacological intervention to treat various inflammatory diseases, including some chronic diseases [24,26], such as osteoarthritis [27,28], metabolic syndrome [29], COPD [30], fibromyalgia [31,32,33], and obesity [34]. Several WBV protocols have been described; however, the mechanisms that underlie the IBR related to the WBV exercise in these clinical statuses are not yet fully clear. Therefore, the aim of this systematic review was to seek a better understanding of the physiological changes generated by WBV exercise on IBR and, consequently, to understand the mechanisms of action that can occur in individuals who perform this type of exercise.

## 2. Materials and Methods

### 2.1. Protocol and Registration

The methods of the analysis, along with the inclusion and exclusion criteria, were specified in advance and documented in a protocol. This work was registered in the International Prospective Register of Systematic Reviews (PROSPERO)—number CRD42020187943.

### 2.2. Research Question

This systematic review aimed to answer the following question: can WBV exert anti-inflammatory action in individuals with different clinical status? The PICOS (P = patients, I = intervention, C = comparison, O = outcomes, S = study design) method was used to define the five major components of the research question [35]: P = individuals who presented with inflammatory conditions; I = WBV; C = comparison of the interventions with and without WBV exercise; O = inflammatory biomarker response (blood markers of inflammation); S = randomized controlled trials (RCTs).

### 2.3. Search Strategy Used to Find the Publications

Three independent reviewers accessed the Physiotherapy Evidence Database (PEDro), Medline/PubMed, Web of Science, SPORTDiscus, Cochrane Library, Cumulative Index to Nursing and Allied Health Literature (CINAHL), and Scopus databases. The search was carried out on 20 May 2020 and repeated on 30 December 2020. The search expressions used in this review were as follows: (1) Medline/PubMed—((((“inflammation” [All Fields]) OR (“inflammatory disease*” [All Fields])) OR (“inflammatory marker*” [All Fields])) OR (“inflammation”[MeSH Terms])) AND (((“whole body vibration” [All Fields]) OR (“wbv”[All Fields])) OR (“vibration” [MeSH Terms])); (2) Scopus—((TITLE-ABS-KEY (“whole body vibration”) OR TITLE-ABS-KEY (wbv))) AND ((TITLE-ABS-KEY (inflammation) OR TITLE-ABS-KEY (“inflammatory disease*”) OR TITLE-ABS-KEY (“inflammatory marker*”))); (3) Web of Science—(TS = (“inflammation”) OR TS = (“inflammatory disease*”) OR TS = (“inflammatory marker*”)) AND (TS = (“whole body vibration”) OR TS = (wbv)); (4) The Cochrane Library—((inflammation):ti,ab,kw OR (“inflammatory disease”):ti,ab,kw OR (“Inflammatory marker”):ti,ab,kw”) AND ((“whole body vibration”):ti,ab,kw OR (“wbv”):ti,ab,kw”); (5) SPORTDiscus and CINAHL—(whole body vibration or whole-body vibration or wbv) AND (inflammation or inflammatory or inflammatory marker). (6) For the PEDro database we used the sentences: (a) whole body vibration or WBV and inflammation; (b) whole body vibration “OR” WBV “AND” inflammatory.

All of the pooled publications were screened according to the inclusion and exclusion criteria.

### 2.4. Inclusion Criteria

All of the publications found in the databases were preliminarily considered to be included in this systematic review. To meet the inclusion criteria, the studies had to include the following: (i) investigation of the effects of WBV exercise on IBR (blood markers of inflammation); (ii) RCTs; and (iii) publications with patients who performed static or dynamic exercises on a VP.

### 2.5. Exclusion Criteria

The exclusion criteria were as follows: (i) review articles; (ii) case reports; (iii) study protocols; and (iv) WBV associated with other therapeutic interventions.

### 2.6. Study Selection and Data Extraction

All publications found on the databases were exported to a data management software platform (Excel), and the duplicates were removed. The review was conducted following four steps. Records were identified through database search and reference screening (Identification), and two reviewers independently examined their titles and abstracts, after which relevant studies were included based on the eligibility criteria (Screening). Relevant full texts were analyzed for eligibility (Eligibility), and all relevant studies were included in the systematic review. The disagreements were resolved by a third reviewer.

The data were extracted from each article and were imported to an Excel spreadsheet containing (1) data regarding study information (i.e., author and year), (2) the clinical status of the participants, (3) aims, (4) characteristics of the participants (e.g., sample size, age, sex) and groups (i.e., WBV or without WBV), (5) inflammation evaluation (blood markers of inflammation), (6) WBV exercise protocols, (7) biomechanical parameters (frequency, peak-to-peak displacement, and peak acceleration), and (8) IBR. Two reviewers independently extracted these data to the spreadsheet. The disagreements were resolved by a third reviewer.

### 2.7. Methodological Quality, Risk of Bias, and Levels of Evidence (LE) of the Selected Papers

The studies were independently appraised by two reviewers and, when there was any disagreement, a third researcher was consulted. The methodological quality was evaluated according the PEDro scale, which tests the methodological quality of clinical trials of physical therapy interventions. On this scale, there are 10 items established based on an “expert consensus” [36]. The publications were classified as “high” (i.e., a score of seven or greater), “fair” (i.e., a score of five to six), or “poor” (i.e., a score of four or below) [37].

The level of evidence of each work was classified according to the National Health and Medical Research Council (NHMRC)’s hierarchy of evidence [38].

The risk of bias of the included studies was evaluated using the Cochrane Collaboration’s risk-of-bias tool. This assesses the internal validity of the trial and the risk of possible bias in different phases of the studies, including (1) random sequence generation, (2) allocation concealment, (3) blinding of participants, (4) personnel and outcome assessment, (5) incomplete outcome measures, (6) selective outcome reporting, and (7) other types of bias. Each item was qualified as low-risk (green), unclear-risk (yellow), or high-risk (red) [39].

## 3. Results

### 3.1. Studies Included

In total, 304 papers (Web of Science = 67, SPORTDiscus = 11, the Cochrane Library = 10, CINAHL = 14, Medline/PubMed = 137, PEDro = 5, Scopus = 60) were initially screened. A total of 119 records were identified as duplicates and removed. Ultimately, only four articles met the inclusion criteria. The selection process is schematized in the PRISMA flowchart [35] (Figure 1).

Table 1 summarizes the publications selected in this systematic review, presenting the characteristics of the WBV protocols, levels of evidence, methodological quality, and the IBR. The included studies showed improvements in the IBR (cytokines, adipokines, and C-reactive protein) in individuals with different clinical conditions (healthy elderly, obese subjects, healthy males, and elderly knee osteoarthritis patients) using several WBV protocols.

### 3.2. Study Population

The selected studies included several populations with different clinical status. One publication analyzed the effects of WBV exercises on knee osteoarthritis [40], one in healthy elderly individuals [19], one in young male students [41], and one in obese individuals [42]. All of the studies aimed to investigate IBR after WBV intervention.

### 3.3. Whole-Body Vibration Exercise Protocols

Almost all of the selected studies used vertical VP, except for one [41], which did not specify the VP movement. The frequency and peak-to-peak displacement (D) of the mechanical vibrations ranged from 12 to 50 Hz and from 2 to 4 mm, respectively, and one publication [41] did not specify the D in mm. One study [41] investigated the acute effects of WBV exercise, while three studies [19,40,42] examined the long-term effects of WBV protocols.

### 3.4. Methodological Quality

All studies were classified as level II according to the NHMRC classification [38]. With regard to the methodological quality (PEDro score), two works [40,41] were considered to be of “high” methodological quality (≥7), one [42] was considered “fair” (5 or 6), and one was considered “poor” (≤4) [19].

### 3.5. Risk of Bias

The risk of bias was evaluated according to the Cochrane Collaboration’s risk-of-bias tool [39]. The detailed assessment of risk of bias is presented in Figure 2 [43]. It was verified that one paper [19] presented a high risk of bias in only one domain (performance bias). One study [41] was classified as having a low risk of bias, while the remaining two were classified as unclear due to insufficient information [40,42].

### 3.6. Inflammation Biomarkers

Two studies analyzed the effects of WBV exercises on the concentrations of C-reactive protein (CRP) [19,42], one analyzed IL-6 [41], one analyzed IL-10 [19], two analyzed tumor necrosis factor alpha (TNF-α) [19,42], one analyzed soluble tumor necrosis factor receptor (sTNFR1, sTNFR2) [40], one analyzed toll-like receptors (TLR2 andTLR4) [19], one analyzed adiponectin [42], and one analyzed leptin [42]. IBRs were analyzed in various populations, as shown in Table 1 and indicated in Figure 3.

## 4. Discussion

The present study investigated the effects of WBV exercise on IBR in individuals with different clinical status. To the best of our knowledge, this is the first systematic review aiming to examine these responses. In this investigation, we included four studies with various protocols and clinical statuses. However, it is important to consider that all studies were classified as level II (NHMRC) [38] (Table 1), and the analysis of methodological quality (PEDro score) (Table 1) indicated that two works were considered “high” (≥7) [40,41], one “fair” (5 or 6) [42], and one “poor” (≤4) [19]. Furthermore, one paper [19] presented a high risk of bias in only one domain (performance bias). One study [41] was classified as having a low risk of bias, while two were classified as unclear [40,42] (Figure 2).

The results of this work show that WBV exercise may be useful in the management of inflammatory conditions by altering the concentrations of some blood markers of inflammation, but this effect is not fully clear yet. However, the IBR promoted by WBV might be justified due to WBV being a kind of physical exercise. It is relevant to highlight that the protocols including WBV can be personalized to subjects with different clinical status. This was also pointed out by Rittweger et al. in 2010 [24] and Wuestefeld et al. in 2020 [25]. Sallam and Laher [44], in a 2016 review of the modulation of oxidative stress and inflammation by physical exercise, suggested that the biological responses related to WBV would depend on various factors, such as the intensity, type, periodicity, and duration of the exercises, along with individual characteristics. Thus, the development of personalized physical exercise programs is essential. In addition, Oroszi et al. in 2020 [20], showed that WBV can promote physiological responses as an indirect effect of the vibratory stimulus; meanwhile Moreira-Marconi et al. in 2020 [45], summarized the hormonal responses in several studies of vibratory stimulus (local and WBV).

### 4.1. C-Reactive Protein (CRP)

CRP has both pro-inflammatory and anti-inflammatory properties; it is synthesized in the liver hepatocytes and—in lesser quantities—by endothelial cells, smooth muscle cells, lymphocytes, macrophages, and adipocytes [46]. It is a blood marker of inflammation that is normally found in chronic systemic inflammation and used in the risk assessment of cardiovascular disease [47]. Physical exercise seems to reduce the CRP levels in individuals with different clinical status. Han et al. [48] showed in a 2019 meta-analysis that physical exercise can be used as a therapy to reverse the low-grade inflammatory state reducing the CRP levels in children and adolescents with overweight or obesity. Fedewa et al. in 2017 [47], suggested that physical exercise and decreases in body mass index are associated with reductions in CRP levels regardless of age or sex. Kohut et al. in 2006 [49], compared aerobic exercises with flexibility/strength exercises and found greater reductions in CRP levels in the former group. This is consistent with the work of Zheng et al. in 2019 [50], which reported that aerobic exercises can have a beneficial effect in reducing the blood markers of inflammation—including CRP levels—in individuals more than 40 years old. Moreover, El-Kader and Al-Jiffri, in 2019 [51], also reported a significant decrease in CRP in obese post-menopausal women who performed aerobic exercise or resistance exercise training. According to the results of this review, the WBV exercise, as along with other modalities of physical exercise, also reduced the CRP levels. Rodriguez-Miguelez et al. in 2015 [19], showed a decrease in the CRP levels in healthy older adults after 8 weeks of WBV exercise (twice a week) using frequencies of 20–35 Hz. On the other hand, a pilot study by Seefried et al. in 2017 [52] observed no significant decline in CRP levels in individuals with end-stage renal disease (i.e., hemodialysis patients) using frequencies of 14–28 Hz along with other exercises, twice weekly for 12 weeks, before or after hemodialysis sessions. Oh et al. in 2019 [18], observed that CRP levels and hepatic stiffness decreased in individuals with nonalcoholic fatty liver disease subjected to WBV exercise (30–50 Hz) twice a week for 6 months, suggesting that a low-intensity WBV program may be considered the best program for patients who have difficulty engaging in exercise.

### 4.2. Cytokines

Cytokines are soluble proteins or glycoproteins that regulate the functions of the immune system, and they can increase (pro-inflammatory) or attenuate (anti-inflammatory) the IBR, making them necessary for the homeostasis of the organism [53]. Changes in the concentrations of cytokines can be balanced with interventions such as physical exercises, including WBV exercise, and this can have beneficial effects on IBR in individuals with various clinical status. Di Giminiani et al. in 2020 [41], found an increase in IL-6 in only one session of WBV (30 or 45 Hz) in young males. Neves et al. in 2014 [54], demonstrated an acute increase in IL-6 after high-intensity physical exercises when compared with low-intensity exercises, while IL-10 showed a greater reduction in response to low-intensity exercises. Cerqueira et al. [12], in a 2020 systematic review, investigated the IBR after different intensities of physical exercise and observed increases in TNF-α and IL-10 only after intense exercise, along with greater increases in IL-6, IL-10, and IL-1β with intense than with moderate exercise. TNF-α is a potent pro-inflammatory cytokine that can perform a variety of biological activities, including inflammation [55]. Hazell et al. in 2014 [56], added WBV exercise (45 Hz) to only one session of the physical exercise (using body mass as the resistance) and found significant increases in IL-1β and IL-6, but with no differences between groups (i.e., with and without WBV). In addition, the IL-10 increased more in healthy males subjected to WBV than in those who engaged in exercise without vibration. Lage et al. in 2018 [57], found higher levels of IL-10 in individuals with COPD after only one session of the WBV exercise (35 Hz) compared with their levels at rest. As IL-10 is an anti-inflammatory cytokine [58], this indicates that WBV exercise can exert significant effects on these anti-inflammatory responses. However, one study compared aerobic exercises with flexibility/strength exercises, measuring IL-6, TNF-α, and IL-18 and finding greater reductions in IL-6 and IL-18 in the aerobic exercises group, whereas TNF-α declined in both groups [49]. Similarly, Zheng et al. in 2019 [50], also found reductions in the TNF-α and IL-6 after aerobic exercises. Oh et al. in 2019 [18], also observed that TNF-α was decreased in individuals with nonalcoholic fatty liver disease after 6 months of WBV exercise (twice a week) at frequencies from 30 to 50 Hz.

sTNFR1 and sTNFR2 are distinct receptors of TNF-α [55]. The expression of these receptors may vary between cell types and tissues. sTNFR1 is expressed on every cell type in the body, while the expression of TNFR2 is limited to endothelial cells, nerve cells, and cells of the immune system [55]. Simão et al. in 2012 [40], found significant reductions in the concentrations of sTNFR1 and sTNFR2 after 12 weeks of WBV (three times per week) using frequencies from 35 to 40 Hz, suggesting that the WBV intervention reduces the IBR in elderly knee osteoarthritis patients. In contrast, Ribeiro et al. in 2018 [31], found a decrease in plasma levels of sTNFR1 and increased levels of sTNFR2 in individuals with fibromyalgia, along with an increase in the sTNFR1 plasma levels in healthy women, after only one session of WBV exercise (40 Hz). This could be explained by the fact that plasma levels of sTNFR1 are already higher in individuals with fibromyalgia at rest, showing a probable attempt to control chronic systemic inflammation [31]. Moreover, Marín et al. in 2011 [59], suggested that the physical exercise load can be increased by using a WBV program or by an improvement of resistance. In addition, it is important to consider the findings described by Rodriguez-Miguelez et al. in 2015 [19], which showed a decrease in TNF-α in healthy older adults after 8 weeks of WBV (twice a week) using frequencies from 20 to 35 Hz, supporting the efficacy of WBV in at least partially counteracting inflammaging.

### 4.3. Adiponectin and Leptin

Adiponectin and leptin are adipokines (or adipocytokines) produced by the adipose tissue. Adipokines present pro- and anti-inflammatory actions as well as acting in satiety mechanisms and body weight maintenance and presenting pro- and anti-nociceptive properties that modulate pain perception [60,61]. Their reduction is correlated with increased levels of pro-inflammatory cytokines such as IL-6, IL-12, IL-18, and TNF-α [61]. Bellia et al. in 2013 [42], observed a significant increase in adiponectin after an intervention consisting of 8 weeks of WBV exercise (three times per week) at a frequency of 30 Hz in obese individuals; however, they found no difference in the leptin levels. Oh et al. in 2019 [18], observed that adiponectin increased in individuals with nonalcoholic fatty liver disease subjected to WBV exercise, together with decreases in the TNF-α and CRP levels that could be attributed to the remarkable improvement of hepatic stiffness as a result of the WBV program. However, Ribeiro et al. in 2018 [31], found decreased plasma levels of adiponectin in individuals with fibromyalgia after one session of WBV exercise (40 Hz). They also found an increase in leptin levels, concluding that a single session of WBV exercise can improve the IBR in patients with fibromyalgia, reaching values close to those seen in healthy women.

### 4.4. Final Considerations

Although the inflammatory process is considered to be a vital defense mechanism for health, acting as a protective response of the immune system [1,2], when innate immunity and inflammation are unregulated it can contribute to the development or maintenance of chronic diseases [1,2,3,4]. In this context, the present study gathered data on IBRs to WBV exercise found in the scientific literature. This review shows that this intervention could be beneficial to pathological processes that involve changes in inflammatory biomarkers. Chronic low-grade inflammation in relation to aging (inflammaging) may be associated with chronic diseases such as cardiovascular disease, metabolic syndrome, cancer, type 2 diabetes, sarcopenia, osteoporosis, Alzheimer’s disease, and frailty, due to a common inflammatory pathogenesis. However, it is not known whether chronic diseases are the source of the inflammatory process or whether the aging process (inflammaging) leads to chronic diseases [62]. Flynn et al. in 2019 [63], reported that inflammaging could be related to sedentary lifestyles, leading to various diseases and metabolic disorders. Moreover, they found that physical exercise had positive influences on chronic diseases associated with inflammaging [63]. The 2019 study by Flynn et al. [63] is consistent with the findings in this review, considering that WBV exercise is also a physical exercise modality [22,23].

The present study details several IBRs and their relationships with conditions such as inflammaging and chronic diseases (i.e., KOA and obesity). Unfortunately, to the best of our knowledge, there are few studies on this subject. This makes the publication of this review relevant, as it shows the importance of conducting more studies investigating IBRs to WBV exercise.

The strength of this work is in proposing the application of WBV exercise as a non-pharmacological intervention for the management of inflammatory diseases. It is highly advantageous for individuals who cannot or choose not to take anti-inflammatory drugs due to their adverse effects. Although the mechanism of action of mechanical vibration on IBRs is not yet clear, this work might stimulate additional investigations of this subject, aiming at the establishment of an optimal and specific WBV protocol for each clinical status.

The findings of this investigation must be interpreted with caution due to important limitations. The first limitation of this systematic review is the inclusion of only RCTs (only four papers). This decision was taken in an attempt to improve the methodological quality of this work by considering only high-quality evidence, since relevant information from non-RCT studies could be added. However, various papers of this kind were included in the Discussion section. The WBV protocols, outcome measures, and populations included in the selected studies were very distinct, and this is a major limitation, making comparisons between the papers and the interpretation of the effects of WBV difficult. Third, the overall methodological quality and the risk of bias vary among the included works.

## 5. Conclusions

In conclusion, considering the findings of the present systematic review, it is possible to conclude that WBV exercise may be useful in the management of inflammation in individuals with different clinical status. However, there is no sufficient evidence to support or refute the notion that WBV exercise could have significant effects on inflammation. Therefore, more investigations should be conducted for verification to increase the scientific evidence related to the results presented in this work.

## Figures and Tables

**Figure 1 ijerph-19-14853-f001:**
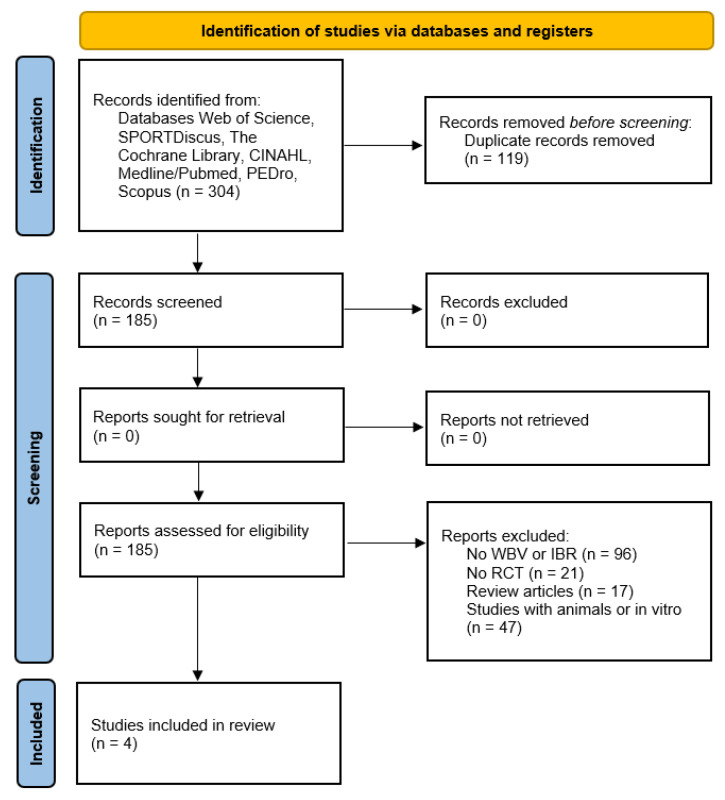
The PRISMA flowchart.

**Figure 2 ijerph-19-14853-f002:**
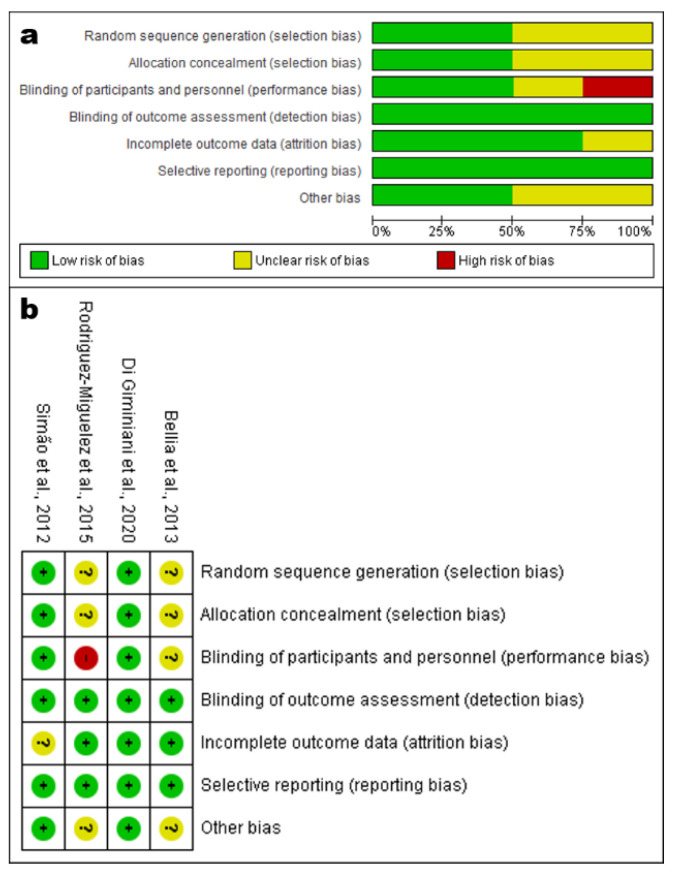
(**a**) Risk-of-bias graph and (**b**) summary indicating the risk of bias of each domain in each study. Green indicates low risk of bias, yellow indicates unclear risk of bias, and red indicates high risk of bias of the studies: Bellia et al., 2013 [40], Rodriguez-Miguelez et al., 2015 [19], Di Giminiani et al., 2020 [41], Simão et al., 2012 [42].

**Figure 3 ijerph-19-14853-f003:**
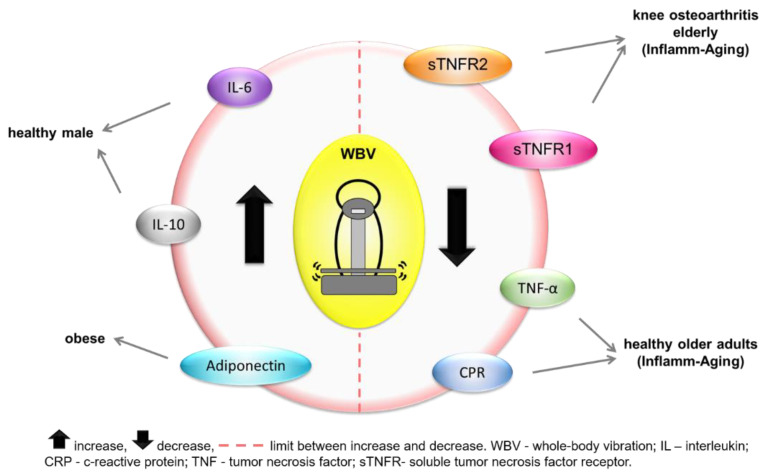
Inflammatory biomarker responses to whole-body vibration intervention in each clinical status.

**Table 1 ijerph-19-14853-t001:** Characteristics of the publications included in the current systematic review.

Author/Year	Clinical Status	Participants/Groups	Inflammation Evaluation	WBV Protocol	Frequency/ Peak-to-Peak Displacement/ apeak	NHMRC/ PEDro Scale	Inflammation Biomarker Responses
Bellia et al., 2013 [40]	Middle-aged obese	N = 34 age 43 ± 6 years (mean ± SD) Male and female Groups: WBV plus hypocaloric diet group or diet alone	High-sensitivity CRP, plasma leptin, adiponectin, and TNF-α	Thrice per week for 8 weeks VP—vertical platform Position—static squatting (110° of knee flexion) Session—10 times of 60 s of vibration and 60 s of rest	*f* = 30 Hz/ *D* = 2 mm a_peak_—no information	L-II 6/10 (fair)	Adiponectin increased in the WBV group compared with the control group.
Rodriguez-Miguelez et al., 2015 [19]	Healthy older adults	N = 38 age 71.04 ± 1.5 years (mean ± SD) Male and female Groups: WBV and control	TLRs (TLR2 and TLR4), TNF-α, IL-10, and CRP	Twice a week for 8 weeks VP—vertical platform Position—static or dynamic exercises: (a) half-squat between 120 and 130° knee angle, (b) deep squat with 90° knee angle, (c) wide-stance squat and calves with a knee angle between 120 and 130° (two sets per exercise mode). Session—30 to 60 s with an interval of 2.5 to 3 min between exercises and 5 min between sets.	*f* = 20 to 35 Hz/ *D* = 4 mm a_peak_—no information	L-II 4/10 (poor)	Plasma levels of CRP and TNF-α decreased in the WBV group from pre- to post-intervention.
Di Giminiani et al., 2020 [41]	Male students	N = 28 Age—WBV: 22.7 ± 0.6 years/control: 22.2 ± 0.8 years (mean ± SD) Male Groups: WBV and control	IL-6	Single session VP—no information Position—isometric half-squat Session—10 series of 1 min and 1 min of rest (4 min of rest after the first 5 series).	*f* = 30 or 45 Hz, *D* = no information a_peak_—individualized determined by EMG	L-II 9/10 (high)	IL-6 increased significantly over time in the WBV group
Simão et al., 2012 [42]	KOA elderly	N = 32 age 75 ± 7.4 years (mean ± SD) Male and female Groups—(1) squat exercises with WBV; (2) squat exercises without WBV; (3) control.	sTNFR1 sTNFR2	Thrice per week 12 weeks VP—vertical platform Position—squat exercise (approximately 10 to 60°of knee flexion) Session—20 to 40 s of the squat with 20 to 25 s of rest (6 to 8 repetitions)	*f* = 35 to 40 Hz,/ *D* = 4 mm/ a_peak_—2.78 to 3.26 g.	L-II 7/10 (high)	In the WBV group, there was reduction on the plasma concentrations of the inflammatory markers sTNFR1 and sTNFR2.

Abbreviations: WBV—whole-body vibration; VP—vibrating platform, f—frequencies; D—peak-to-peak displacement; apeak—peak acceleration; NHMRC—levels of evidence according to the National Health and Medical Research Council; HIIT—high-intensity interval training, SD—standard deviation; CRP—C-reactive protein; IL—interleukin; TNF—tumor necrosis factor; sTNFR—soluble tumor necrosis factor receptor, KOA—knee osteoarthritis, PEDro score—(a) “high” methodological quality ≥ 7, (b) “fair” methodological quality = 5 or 6, (c) “poor” methodological quality ≤ 4.

## Data Availability

Not applicable.

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
