# Peer review of "Inflammatory Biomarker Responses to Whole-Body Vibration in Subjects with Different Clinical Status: A Systematic Review"

_ijerph, 2022, doi:10.3390/ijerph192214853_

Round 1

Reviewer 1 Report

The review is a good start to evaluate the effect of whole body vibration on IBR, it is worth publishing but the following queries must be clarified in the manuscript before accepting for the publication. 

1. Why the following is kept as part of the abstract? this may be stated in the methodology. 

"A Systematic Review according........and Scopus databases."

2. in the last para of the introduction, the research gap and need for WBV is described in the light of literature review. Although the clarity of the objective of the present review is missing, which could have set the live for this systematic review. 

3. Why the following were exclusion criteria?

 (i) review articles; and (ii) case reports; 

4. in sub section it is written " Of these, 4 articles met the inclusion criteria. ". Then will it be worth to consider a review publication only based on four articles? 

5. In discussion, mostly the reproduction of all the four studies selected were seen. Although, it is important to develop a line of action to define the levels of WBV in terms of Frequency and 'D', which may have lesser values of IBR/non inflammatory benefits to the patients of diabetes, post trauma etc.

Author Response

Responses to Comments and Suggestions to Authors - Reviewer 1

The review is a good start to evaluate the effect of whole body vibration on IBR, it is worth publishing but the following queries must be clarified in the manuscript before accepting for the publication.

  1. Why the following is kept as part of the abstract? this may be stated in the methodology.

"A Systematic Review according........and Scopus databases."

R: Thank you. We agree with your consideration and exclude this information from the abstract.

  1. in the last para of the introduction, the research gap and need for WBV is described in the light of literature review. Although the clarity of the objective of the present review is missing, which could have set the live for this systematic review.

R: Thank you. We modified the last paragraph for a better understanding of the purpose and relevance of the present study.

“WBV exercise is produced in individuals who are in contact with the base of a vibrating platform (VP), that transmits mechanical vibration for them [24,25]. WBV exercise has been suggested as a non-pharmacological intervention to treat different inflammatory diseases including some chronic disease [24,26], such as osteoarthritis [27,28], metabolic syndrome [29], COPD [30], fibromyalgia [31–33] and obesity [34]. Different WBV protocols have been described, however, the mechanisms that underline the IBR in these clinical statuses related to the WBV exercise are not fully clear yet. Therefore, the aim of this systematic review was to seek a better understanding of the physiological changes generated by WBV exercise on IBR and, consequently, to understand the mechanisms of action that can occur in individuals that perform this type of exercise.”

  1. Why the following were exclusion criteria?

(i) review articles; and (ii) case reports;

R: Thank you. But these types of study (review articles and case reports) were excluded because the present study is a systematic review based on randomized clinical trials (RCT), as described in the inclusion criteria. We comented about this in the end of the discution, when writen about the limitations of this study. "The first limitation of this systematic review is the inclusion of only RCT (only four papers). This was done in the attempt of improving the methodological quality of this work compiling only high-quality evidence, since relevant information from non-RCT studies could be added."

  1. in sub section it is written " Of these, 4 articles met the inclusion criteria. ". Then will it be worth to consider a review publication only based on four articles?

R: Thank you. We agree with your consideration. For this reason we included the following considerations in our conclusion: “...However, there is no sufficient evi-dence to support or refute the notion that WBV exercise could have significant effects on inflammation. Therefore, more investigations might be conducted to verify to in-crease the scientific evidence related to results presented in this work.” Moreover, we comented about this in the end of the discution, when writen about the limitations of this study. "The findings of this investigation must be interpreted with caution due important limitations. The first limitation of this systematic review is the inclusion of only RCT (only four papers). This was done in the attempt of improving the methodological quality of this work compiling only high-quality evidence, since relevant information from non-RCT studies could be added."

  1. In discussion, mostly the reproduction of all the four studies selected were seen. Although, it is important to develop a line of action to define the levels of WBV in terms of Frequency and 'D', which may have lesser values of IBR/non inflammatory benefits to the patients of diabetes, post trauma etc.

R: Thank you. Our study aimed to investigate the mechanism of action of mechanical vibration on blood inflammatory markers. In the literature, studies involving WBV exercise generally show physical-functional results. For this reason, we focused research on inflammatory processes. In addition, we inform as a limitation of our review the impossibility of an ideal protocol due to the diversity of population and protocols used in the publications included. However, we have added a paragraph relating our findings to 2 articles on Inflammaging and Chronic Diseases.

Reviewer 2 Report

Congratulations on your study, I think it’s really interesting. I propose some advices to improve your study. 

Line 103-104: could authors include in the text?

In the proposed question “Can WBV have anti-inflammatory action in individuals with different clinical status?” Authors exposed different clinical status, however, the keywords didn’t include terms that could include multiple health conditions, such as the term“chronic diseases”. Maybe the use of more keywords can increase the number of studies included. A systematic review with only 4 studies is poor to get a strong conclusion. 

Author Response

Responses to Comments and Suggestions to Authors - Reviewer 2

Congratulations on your study, I think it’s really interesting. I propose some advices to improve your study.

Line 103-104: could authors include in the text?

R: Thank you. We agree that it is not necessary. We remeved it.

In the proposed question “Can WBV have anti-inflammatory action in individuals with different clinical status?” Authors exposed different clinical status, however, the keywords didn’t include terms that could include multiple health conditions, such as the term“chronic diseases”. Maybe the use of more keywords can increase the number of studies included. A systematic review with only 4 studies is poor to get a strong conclusion.

R: Thank you. Our study aimed to investigate the mechanism of action of mechanical vibration on inflammatory blood markers. In the literature, studies involving WBV exercise usually show physical-functional outcomes. For this reason, we focused research on inflammatory processes regardless of the clinical condition addressed. So all studies that contained inflammatory responses were included, including the chronic diseases. Unfortunately, there are few studies on the subject and therefore we include this information both in the study limitation and in the conclusion. This makes the publication of this review even more relevant, as it shows the importance of having more studies investigating the inflammatory biomarkers responses to WBV exercise.